# The New Paradigm: The Role of Proteins and Triggers in the Evolution of Allergic Asthma

**DOI:** 10.3390/ijms25115747

**Published:** 2024-05-25

**Authors:** Ilaria Baglivo, Vitaliano Nicola Quaranta, Silvano Dragonieri, Stefania Colantuono, Francesco Menzella, David Selvaggio, Giovanna Elisiana Carpagnano, Cristiano Caruso

**Affiliations:** 1Centro Malattie Apparato Digerente (CEMAD) Digestive Disease Center, Fondazione Policlinico Universitario “A. Gemelli” IRCCS, Università Cattolica del Sacro Cuore, 00168 Roma, Italy; 2Department of Basic Medical Sciences, Neuroscience and Sense Organs, Section of Respiratory Disease, University “Aldo Moro” of Bari, 70121 Bari, Italysilvano.dragonieri@uniba.it (S.D.);; 3Unità Operativa Semplice Dipartimentale Day Hospital (UOSD DH) Medicina Interna e Malattie dell’Apparato Digerente, Fondazione Policlinico Universitario “A. Gemelli” IRCCS, Università Cattolica del Sacro Cuore, 00168 Roma, Italy; 4Pulmonology Unit, S. Valentino Hospital-AULSS2 Marca Trevigiana, 31100 Treviso, Italy; 5UOS di Malattie dell’Apparato Respiratorio Ospedale Cristo Re, 00167 Roma, Italy

**Keywords:** allergic asthma, epithelial barrier damage, protease allergen, type 2 inflammation, airway remodeling

## Abstract

Epithelial barrier damage plays a central role in the development and maintenance of allergic inflammation. Rises in the epithelial barrier permeability of airways alter tissue homeostasis and allow the penetration of allergens and other external agents. Different factors contribute to barrier impairment, such as eosinophilic infiltration and allergen protease action—eosinophilic cationic proteins’ effects and allergens’ proteolytic activity both contribute significantly to epithelial damage. In the airways, allergen proteases degrade the epithelial junctional proteins, allowing allergen penetration and its uptake by dendritic cells. This increase in allergen–immune system interaction induces the release of alarmins and the activation of type 2 inflammatory pathways, causing or worsening the main symptoms at the skin, bowel, and respiratory levels. We aim to highlight the molecular mechanisms underlying allergenic protease-induced epithelial barrier damage and the role of immune response in allergic asthma onset, maintenance, and progression. Moreover, we will explore potential clinical and radiological biomarkers of airway remodeling in allergic asthma patients.

## 1. Introduction

Epithelial barrier damage is a crucial feature of inflammatory allergic diseases. The epithelium plays a doubly protective role—on a mechanical level, maintaining skin and mucosal barrier integrity; and on an immunological one, through the action of a rich set of molecules that ensure the immune tolerance.

The epithelial barrier is a structured entity in which cell–cell adhesion complexes ensure integrity and effectiveness [1].

Tight Junctions (TJs) include transmembrane proteins of the claudin family, occludin, tricellulin, junctional adhesion molecules, and cytoplasmic proteins (such as the Zonula Occludens (ZO)-1 ZO-2, ZO-3).

Adherens junctions are composed of cadherin–catenin complexes and they act as key regulators of paracellular permeability [2].

Desmosomes provide mechanical stability and hemidesmosomes contribute to epithelial layer–basal membrane attachment.

In normal conditions, a functional physical barrier contributes to the regulation of epithelial permeability, cell proliferation, and differentiation. The loss of barrier integrity increases the exposure to environmental, allergic, and toxic substances, decreasing immune tolerance and inducing the activation of different pro-inflammatory pathways.

Epithelial barrier damage characterizes different inflammatory diseases such as asthma, Chronic Rhinosinusitis with Nasal Polyps (CRSwNP), Eosinophilic Esophagitis (EoE), and Atopic Dermatitis (AD); different mechanisms could contribute to the barrier’s dysfunction.

In allergic diseases, the disruption of the epithelial barrier is associated with TJ defects and with reductions in the numbers of adherence junctions and desmosomes [3,4,5,6].

Zonulin is a regulator of epithelial and endothelial barrier function. It regulates intestinal permeability by disrupting TJs. Defective epithelial barrier function is a hallmark of airway inflammation in asthma [3].

Both environmental and genetic factors are involved in barrier damage [7,8].

Several susceptibility genes have been associated with epithelial barrier differentiation and homeostasis [9,10]. Structural airway remodeling signs have been found in children genetically predisposed to asthma [11,12,13].

Moreover, the environmental context, including the actions of viruses [14], pollutants [15], cigarette smoke [16], and allergens, plays a central role in epithelial injury. Moreover, industrialization and the consumption of highly processed food can contribute to altering the gut microbiota and the intestinal barrier, thus increasing susceptibility to allergic sensitization [17].

Several allergens and allergen components, such as house dust mite (HDM) Der p 1, have shown the ability to disrupt the TJs through both direct and indirect proteolytic activity [18]. 

Increases in epithelium permeability lead to Type 2 (T2) cytokine production and eosinophil activation and proliferation in the airways. Both T2 cytokines and eosinophil mediators interfere with TJs, contributing to the lack of barrier response. In asthma, the inflammatory processes start from the release of alarmins; the intensity of the cytokine release correlates with the clinical symptoms, the disease severity, and the airway remodeling process [19] (Figure 1).

We aim to explore the molecular mechanisms underlying allergenic protease-induced damage in allergic asthma onset, maintenance, and progression. Moreover, we will highlight the roles of both epithelial barrier dysfunction and immune response in airway remodeling, explore potential clinical and radiological biomarkers, and examine different therapeutic options.

## 2. Molecular Mechanisms in Allergic Asthma: The Allergen Proteases

Allergen proteases are proteolytic enzymes that have a primary role in the pathogenesis of respiratory allergies, facilitating the allergen–host interactions and promoting the development of allergic sensitization. 

Allergens and pathogens with proteolytic activity can intrinsically overcome the host’s tolerance, activating various immunological pathways. Proteases enhance antigen-presenting cells in airways, inducing specific Immunoglobulin-E (IgE) production, eosinophil recruitment, and inflammatory mediator release in airways, skin, and other barrier tissues. 

Proteases can be released as enzymatically inactive zymogens, requiring additional adjuvants for the activation, or they can show independent proteolytic activity, as HDM does [20,21,22,23]. 

The HDM major protease Der p 1 is a cysteine protease which has shown self-maturation capacity in acidic ambience [24]; moreover, it has been demonstrated to promote the maturation of other HDM proteases, such as the HDM serine proteases Der p 3, Der p 6, and Der p 9, which require enzymatic activation [25].

Beyond activating factors, specific and non-specific protease inhibitors also participate in the regulation of proteolytic activity, ensuring the maintenance of tissue homeostasis [23].

Although HDM Der p 1 was the first allergen protease to be characterized [26], more recently, different HDM proteases—as well as others produced by cockroaches [27], fungi [28], and plants—have also been described [29].

Based on the catalysis mechanism, according to the locations of their cleavage sites and the natures of their active site residues, proteases have been classified into five classes: aspartic, metallo, cysteine, serine, and threonine proteases [30].

Although all five classes of proteases are found in the human genome [31], only aspartic, cysteine, and serine proteases have been identified as allergens [32].

Most cysteine proteases share structural homologies with Der p 1, having cysteine-histidine-asparagine as an active site residue, while serine protease allergens are structurally similar to trypsin, with serine-histidine-aspartic acid as their active site residue.
*House Dust Mite*

Mite allergen proteases include the papain-like cysteine proteases from group 1 (Der p 1, Der f 1); those from group 2 (Der p 2, Der f 2), which are lipid-binding proteins causing sensitization in more than 90% of mite-allergic patients [33]; and the proteases from group 3 (Der p 3, Der f 3), group 6 (Der p 6, Der f 6), and group 9 (Der p 9), which are serine proteases with trypsin, chymotrypsin, and collagenase activity, respectively.

Der p 1 can damage the bronchial epithelial barrier by degrading the endogenous protease inhibitors, such as lung α1-antitrypsin and elafin [34]; moreover, both Der p 1 and Der f 1 can degrade the airway surfactant proteins Sps-A and Sps-D [34,35].

Proteases’ actions favor allergen penetration and, subsequently, allergen–immune cell contact. Der p 1 interacts with multiple molecules involved in the control of IgE synthesis [36,37].
*Fungi*

Fungal proteases are strong activators of T2 inflammation and play a major role in epithelial damage mechanisms. Fungal proteases, as well as HDM ones, can act as adjuvants of endogenous proteases and degraders of protease inhibitors.

The major fungal proteases are serine proteases, such as those of *Cladosporium cladosporioides* [38], *Penicillium*, and *Aspergillus* species [39]; however, aspartate proteases have been described in *Alternaria alternata* [40].

*Aspergillus fumigatus*, *Alternaria alternata*, and *Cladosporium herbarum* proteases induce morphologic changes and cell desquamation in the cultured airway epithelial cells, favoring the release of proinflammatory cytokines [41].

In particular, *A. alternata* proteases induce intense eosinophilic activation: the addition of aspartate protease inhibitors to *A. alternata* extract has shown to attenuate the eosinophils’ response [40].

Fungal proteases, as well as Der p 1, interact with the kinin system, the coagulation cascade, and the fibrinolytic mechanism. The release of fibrinogen cleavage products induced by prothrombinase activity stimulates the innate immune response through the activation of Toll-Like Receptor-4 (TLR4) [42,43]. Moreover, thrombin is involved in different signaling pathways inducing the IgE-independent cytokine production [44].
*Cockroaches*

The only cockroach allergen showing proteolytic activity is the *Periplaneta americana* serine protease Per a 10 [45], which induces both self-activation and adjuvant effects in inactive proteases. Although *Blattella germanica* extract is rich in proteases that show direct proinflammatory effects on the airway epithelial tissue, none of them have shown direct proteolytic activity [46]. An aerosolized cockroach extract has been shown to induce airway eosinophilic inflammation in animal models [47,48].
*Foods*

Food components, such as melon, kiwi, papaya, and other fruits, could induce allergic sensitization and have shown serine and cysteine proteolytic actions. Papain, a papaya-derived allergen, belongs to the same family of cysteine proteases as the HDM major group 1 allergens. Papain has been shown to activate both innate and Th2 immune responses [49], inducing alarmin release [50,51], the activation of mouse basophils in vitro [52], and lung eosinophilia in mice [53,54].
*Pollen*

Pollen proteolytic activity has been attributed to both allergenic and non-allergenic cysteine, serine, and metalloprotease [55]. IgE-reactive cysteine proteases are present on the coats of *Cynodon dactylon*, *Sorghum halepense*, and *Phleum pratense* pollen [56]. In *Ambrosia artemisiifolia* pollen, the allergenic cysteine protease Amb a 11 has been isolated [29]. *Betula verrucosa* contains proteases potentially homologous to Der p 1 [57].

Taken together, allergen proteases alter epithelial cells and cell junctions, promoting external agents’ penetration and the activation of different inflammatory pathways.

### The Allergen Proteases in Epithelial Barrier Damage and Inflammatory Signals

The airways, skin, and gastrointestinal tract are the main tissues involved in allergen protease-driven epithelial barrier damage [58,59].

In airway epithelia, HDM exerts proteolytic activity through both direct and indirect mechanisms, involving direct occludin and claudin degradation [60,61] and the Protease-Activated Receptors’ (PARs) activation, respectively [23,60,62]. A primary role has been attributed to Der p 1, although HDM serine peptidases have also shown the ability to damage the epithelium [23].

It has been observed that Der p 1 could cause the detachment of bronchial epithelial cells [18,63,64].

The main role of Der p 1 is confirmed by the substantial inhibition of HDM activity if Der p 1-selective inhibition is provided [23].

Similar mechanisms are exploited by the fungal serine proteases Pen c 13 and Asp f 13 and the cockroach protease Per a 10 [65,66]. Moreover, in human bronchial epithelial cells, the fungal protease Pen c 13 has been shown to downregulate the expression of CD44, which is involved in epithelial repair mechanisms [67].

In the gastrointestinal tract, it has been demonstrated that allergen proteases, such as the kiwi fruit actinidin (Act d 1), affect occludin and ZO-1, increasing intestinal permeability [68].

Damaged epithelia are easily crossed by allergens and other external agents that directly reach Dendritic Cells (DCs), inducing modifications in cell surface receptors and unbalancing the immune response toward a T2 phenotype.

Active proteases induce the proteolytic cleavage of CD40, which results in a reduction of the Type 1 (T1) inflammatory mediators’ release, along with weak IL-12 production and, in contrast, increased levels of IL-4 and IL-13 [69].

Der p 1 can not only induce the soluble CD40 directly from DCs’ surfaces [69], but it can also cleave the DC-SIGN (CD209) [70], a receptor involved in T1 cell differentiation [71]. Moreover, Der p 1 upregulates the expression of CD86, favoring the expression, in DCs, of chemokines involved in T2 response [72].

Der p 1, as well as Per a 10, has been shown to modulate both T and B cells through the direct cleavage of CD25 (the alpha chain of the IL-2 Receptor) and CD23 (the low-affinity receptor for IgE) [73], resulting in a lower release of IL-12 and Interferon (INF), increasing T2 cytokine levels and IgE synthesis.

In addition, the CD23 cleavage could further increase IgE synthesis, disrupting the negative feedback between the membrane-bound form of IgE Receptor and the IgE production [74] (Figure 2).

Allergen proteases could activate the mast cells through both IgE-mediated and non- IgE-mediated mechanisms. Non-IgE-mediated mast cells’ activation involves the cell surface PARs [75,76].

PAR-1, -3, and -4 are activated by thrombin, while PAR-2 is activated by trypsin, which shares molecular features with allergic proteases [75].

Epithelial cells, mast cells, basophils, eosinophil, and other cellular types are all involved in the PAR-2-mediated allergen protease response, as observed after Der p 1, Der p 3, and Der p 9 exposure [77,78].

In airway epithelial cells, PAR-2 activation induces cellular morphologic changes, cell desquamation, and the release of cytokines, growth factors, and prostanoids [50].

The inflammatory environment enhances PAR expression [79]; this has been demonstrated in comparisons between asthma patients’ bronchial epithelium biopsies and control biopsies [80].

Moreover, allergens can upregulate PAR-2 levels on pulmonary and bone marrow-derived myeloid Dendritic Cells (mDCs) [81]. An overexpression of PAR-2 and PAR-3 mRNAs has been described in nasal polyp epithelial cells stimulated with *Aspergillus*, *Alternaria*, and *Cladosporium* [82].

PAR-2 favors the recruitment of the alveolar macrophages [83] neutrophil and eosinophil. In particular, the role of PAR-2 in eosinophil’s activation has been confirmed by the inhibition of eosinophilic response—stimulated with exposure to the cell-free extract of *A. alternata*—in the presence of protease ligands and PAR-2 antagonist peptide [40].

Active Der p 1, Der f 1, or papain lead to superoxide anion production through direct eosinophil activation [84].

Basophils exposed to proteolytically active Der p 1 or papain secrete Thymic Stromal Lymphopoietin (TSLP) and IL-4 in an IgE-independent way [20,85]; the specific mechanism is unknown, although nociceptive primary sensory neurons, namely Mas-related G-protein-coupled receptors (Mrgprs), seem to be involved [86].

The early role of type 2 Innate Lymphoid Cells (ILC2) in T2 immune response is well-known. Damaged epithelium produces alarmines, such as TSLP, IL-25, and IL-33, which activate ILC2 to produce large amounts of IL-4, IL-5, and IL-13, promoting Th2 differentiation. Th2 cells contribute to T2 cytokines’ release and mediate allergen-specific IgE production [87].

In allergic patients, Der p 1 and *Aspergillus* have shown to induce ILC2 recruitment and activation [54,88].

A contribution to tissue injury is provided by alterations to the protease/anti-protease balance.

As mentioned above, the physiological cellular protective function is ensured by the activity of anti-proteases, such as 1-antitrypsin, elafin, and Secretory Leukocyte Proteinase Inhibitor (SLPI). The protease/anti-protease balance is critical for protecting lung tissue, since the loss of their homeostasis is a feature of emphysema and asthma [34].

Beyond the degrading effects of allergen proteases (such as papain, Der p 1, cat skin) [34], genetic factors could cause the loss of protease inhibitors’ expressions, contributing to exogenous damage [89,90].

Mechanical and immunological epithelial barrier dysfunction induces molecular, cellular, and tissue modifications that are features of allergic asthma.

In this context, chronic inflammation predisposes patients to the occurrence (and enhancement) of airway remodeling and asthma exacerbation.

## 3. Molecular Mechanisms in Allergic Asthma: Airway Remodeling

### 3.1. Biomarkers of Airway Remodeling

Asthma-related airway remodeling includes structural changes like sub-epithelial fibrosis, thicker Airway Smooth Muscle (ASM), mucous gland hyperplasia, angiogenesis, and damaged epithelial layers, resulting in stiffer airway walls [91]. Such remodeling significantly contributes to persistent symptoms and severity in severe asthma cases [92]. Notably, airway remodeling can begin early, even before asthma diagnosis, as observed in preschool children with confirmed wheezing [93]. The identification of potential biomarkers could aid in detecting early signs of remodeling.

#### 3.1.1. Epithelial Remodeling

Asthma-induced airway epithelium remodeling involves epithelial cell deterioration or loss, decreased ciliated cells, and increased goblet cells [94]. Epithelial–Mesenchymal Transition (EMT) is crucial in this process, driven by Transforming Growth Factor beta (TGF-β), leading to epithelial cells transforming into mesenchymal cells [95]. Markers include reduced E-cadherin and increased N-cadherin [96]. The IL-33/CD146 axis influences EMT in asthma, with HDM extract boosting IL-33 and CD146. Epithelial cell interactions with the immune system may involve Extracellular Vesicles (EVs)—with altered microRNA (miRNA) contents in response to stress or activation—playing a role in asthma development [97]. The communication between lung epithelial cells and the immune system may involve EVs carrying miRNAs. These miRNAs, which change due to cellular stress or activation [98], are crucial in asthma [99,100,101], showing different levels in asthma patients’ bronchoalveolar lavage fluid compared to that of healthy individuals [102]. Research indicates that specific miRNAs in EVs from airway epithelium, like miR-34a, miR-92b, and miR-210, could be key in initiating Th2 responses and asthma development [103].

#### 3.1.2. Reticular Basement Membrane Thickening

Research links Reticular Basement Membrane (RBM) thickening in asthma to gene expressions influencing airway growth and fibrosis, affecting various physiological processes [104]. Identifying specific fibrocytes in Bronchoalveolar Lavage Fluid (BALF) marked by CD34/CD45RO/α-SMA/procollagen I, indicative of basement membrane thickening, suggests a role in mild asthma’s airway remodeling, with future non-invasive detection possibilities [105]. A study on severe asthma identified galectin-3 as a biomarker in omalizumab-treated patients, distinguishing responders by their protein profiles related to smooth muscle and extracellular matrix [106].

#### 3.1.3. Subepithelial Fibrosis

TGFβ plays a crucial role in asthma by transforming airway fibroblasts into myofibroblasts, leading to subepithelial fibrosis [107]. The severity of fibrosis correlates with TGF-β1 mRNA levels in bronchial biopsies [108], and elevated αvβ8 integrins in asthma indicate their potential as biomarkers [109]. Periostin, associated with IL-4 and IL-13, impacts fibrosis and inflammation, marking the efficacy of Th2 antagonists [110]. Follistatin-like 1 (FSTL1)-induced autophagy may promote epithelial–mesenchymal transition, suggesting its potential for new asthma treatments [111].

#### 3.1.4. Airway Smooth Muscle

Many ASM cell mitogens are involved in asthma, such as Platelet-Derived Growth Factor (PDGF), TGF-β, Epidermal Growth Factor (EGF), Heparin-Binding EGF, and Vascular Endothelial Growth Factor (VEGF) [112].

ASM, histologically assessed by endobronchial biopsies, has been recognized as a valuable biomarker in phenotyping airway diseases, especially in the context of personalized medicine [113].

TGF-β stands out as a potential biomarker for this mechanism, as it becomes activated when ASM cells and the airways contract. TGF-β is known as a cytokine that promotes remodeling processes [114]. Additionally, pharmacological means to inhibit Transient Receptor Potential Vanilloid-1 (TRPV1), a factor that can influence the tone of ASM and effectively mitigate airway remodeling in living organisms, are promising [115].

Recent studies highlight the absence of a complete molecular marker system for ASM cells (ASMCs), yet remain hopeful for future developments. It has been discovered that Myosin Heavy chain 11 (MYH11) serves as a marker for mature SMCs, and Transgelin (TAGLN) indicates early SMC differentiation. This suggests the possibility of using various molecular markers or their combinations to identify the properties and origins of increased ASMCs in asthma-related airway remodeling, depending on the stage of differentiation and research requirements [116].

DNA methylation changes in severe asthma, particularly in regard to ASMCs, illuminating disease mechanisms. Asthma shows reduced methylation in the Phosphodiesterase 4D (PDE4D) promoter area, impacting ASMCs’ proliferation [117]. These patterns relate to asthma’s severity and correlate with gene and miRNA changes, affecting ASMC function. This suggests the potential use of demethylating agents in severe asthma treatment [118]. Integrins, which are crucial in ASM contraction and remodeling, mediate ASM and extracellular matrix interactions. Fibronectin-binding α5β1, α2β1, and α9β1 integrins could be therapeutic targets [112].

#### 3.1.5. Mucus

In asthma, the hypersecretion of mucins MUC5AC and MUC5B by goblet cells contributes to airway remodeling. While MUC5B performs essential homeostatic roles, targeting MUC5AC secretion could be a potential therapeutic strategy [119,120].

#### 3.1.6. Vasculature

Many studies have observed changes in the bronchial vascular network in asthma, including increased blood vessel number, size, and density; vascular leakage; and plasma engorgement. This neovascularization, a key element of airway remodeling, has uncertain effects on bronchial walls and lung function. Contributing factors include extracellular matrix alterations and growth factor dysregulation [121]. VEGF, a key stimulator of endothelial cell growth and vascular permeability, is elevated in asthma, and specific integrins like αvβ3 and αvβ5 play vital roles in blood vessel development [112].

### 3.2. Airway Remodeling: Radiological Pathways and Key Points

High-resolution Computed Tomography (HRCT) is crucial in identifying radiological markers in asthma, revealing both static and dynamic airway changes as small as 1 mm in diameter [122,123]. In patients with stable asthma who undergo computer CT scans, three primary measurements are acknowledged as efficient in assessing airway remodeling: the percentage of bronchial Wall Thickness (WT%), the Bronchial-to-Arterial diameter ratio (BA ratio), and the level of Airway Collapsibility (AC) during both inhalation and exhalation. This evaluation of airway remodeling relies on the post-bronchodilator [124].

In a significant study, about 80% of severe asthma patients showed chest CT abnormalities, highlighting CT’s value in assessing this condition [125].

Hartley et al. discovered a negative correlation between Wall Area percentage (WA%) and Forced Expiratory Volume in 1 s (FEV1) in non-smoking asthma patients. This indicates that WA% is a crucial marker for assessing airway remodeling in severe asthma, highlighting the relationship between airway WT and lung function impairment [126].

Quantitative CT (QCT) scans are effective biomarkers for airway remodeling, significantly enhancing the precise analysis and understanding of severe asthma [92,127,128,129]. QCT biomarkers like WT%, WA%, and air trapping (measured through low-attenuation area) are higher in asthma patients compared to controls [130] and are particularly elevated in severe cases [131]. These QCT measures correlate closely with asthma severity and histological findings, making them effective for both studying and monitoring asthma [132].

#### 3.2.1. Radiological Indicators for Assessing Severity, Early Identification, and Involvement of Small Airways

Bronchial WT (BWT) and emphysema are more common in patients with severe asthma compared to those with mild asthma [125,133,134,135]. However, other studies have not found a correlation with the severity of asthma [133,136,137]. WT% is a meaningful radiological marker in assessing lung function changes in asthma. In the Severe Asthma Research Program (SARP) study, which focused on never smokers and ex-smokers with a history of less than 10 packs per year, it was found that the WT% was notably higher in asthmatic patients who experienced a significant decline in lung function over a three-year period compared to those whose lung function remained normal or improved [138]. Similarly, in HRCT imaging, patients with lower bronchodilator-responsive FEV1 had twice the WT compared to those with normal FEV1 (about 90% predicted), underscoring a significant relationship between increased BWT and diminished lung function [139].

Emphysema-linked changes notably impact lung function in asthma patients regardless of smoking habits [140,141], indicating the permanent nature of persistent airway obstruction in severe asthma patients, particularly in those with a significant reduction in baseline bronchodilator-responsive FEV1. In a study by Kim YH et al., emphysema scores were four times higher in the Tr5 group compared to the Tr4 group, a trend also observed among non-smokers [139]. Research has shown that 15% to 39% of people with asthma, including non-smokers, experience these changes [142]. CT-measured air trapping in asthma patients is linked to the severity of their asthma and an increased likelihood of experiencing severe exacerbations [143]. Patients undergoing three months of inhaled corticosteroid therapy showed reduced air trapping in their CT scans [144]. Additionally, a study by Haldar et al. on 26 patients with severe, eosinophilic, refractory asthma revealed that one-year treatment with mepolizumab, an anti-IL5 monoclonal antibody, significantly lowered average wall area compared to a placebo [145].

Delta Lumen, defined as the percent change in airway lumen area between Functional Reserve Capacity (FRC) and Total Lung Capacity (TLC), is a new metric in a study of 152 asthma patients. It negatively correlates with WT% and low-attenuation area, especially in severe cases like refractory asthma requiring systemic corticosteroids or hospitalization due to exacerbation. This suggests that a reduced Delta Lumen, as measured by QCT, could be a useful biomarker for identifying severe, unstable asthma [146].

The more pronounced thickening of airway walls observed in HRCT images can act as an early indicator of airway remodeling in asthma cases, even when lung function tests like FEV1 are normal [147].

In persistent asthma, the tBTW is linked to increased resistance and reactance in peripheral airways, a higher frequency of severe exacerbations, and the presence of nasal polyposis [148]. QCT has shown a strong correlation between bronchial lumen area and inner diameter through lung function tests in a study of 83 long-term asthma patients. Notably, these measures were reduced from the seventh to the ninth bronchial generations, indicating airway remodeling predominantly in medium and small airways [149].

#### 3.2.2. Radiological Pathways in Patient Phenotyping

Various attempts have been made to phenotype asthmatic patients through radiological pathways.

The WA% is significantly higher in asthma patients than in those with Eosinophilic Bronchitis (EB), with a difference of 72 (3.1) % versus 54 (2.1) %. This suggests that in asthma, increased WA% might play a more critical role in airway hyperresponsiveness than factors like air trapping or centrilobular prominence, which are typically considered to affect it more. In contrast, the WA% in EB patients is not as prominently different [150].

In a cluster-based study of asthma patients, clusters with a higher bronchial wall area in their right upper lobe’s bronchus, as assessed by CT, were associated with elevated sputum neutrophil levels [151]. In 2014, Gupta et al., through cluster analysis, identified three novel asthma phenotypes with unique clinical and radiological characteristics. Cluster 1 showed an increase in the lumen volume and a decrease in the percentage wall volume of the right upper lobe apical segmental bronchus. In contrast, Cluster 3 had the smallest lumen volume, but the highest percentage wall volume in the same bronchus. Cluster 2, however, displayed an absence of proximal airway remodeling. These findings suggest distinct structural changes in the airways of different asthma phenotypes [134].

In an HRCT study of 109 untreated asthma patients, key findings included airway remodeling, bronchiectasis, and mucus plugs, which were more pronounced than in healthy individuals. A notable inverse relationship existed between airway WT and mid-expiratory flow [152].

In a study of 61 asthmatic patients, four QCT-based clusters were identified, differing in asthma severity and lung function decline over five years. Cluster C1 consisted of non-severe asthmatic patients with increased wall thickness; C2 had a mix of severe and non-severe cases with limited bronchodilator response; and C3 and C4 included severe asthmatic patients, with C3 focusing on severe allergic asthma without small airway disease, and C4 on ex-smokers with significant small airway disease and emphysema [153]. Kim S. et al. categorized asthma airway remodeling into three types: Large Airway Involvement (LA), Small Airway Involvement (SA), and Normal/Near-Normal (NN). In their study of 91 severe asthma patients, 81.3% showed bronchial thickening and bronchiectasis, 6.6% had small airway remodeling associated with airflow obstruction and smoking, and 26% displayed no significant remodeling and required fewer oral corticosteroids [154].

The radiological markers and biomarkers of airway remodeling are summarized in Table 1 and Table 2.

## 4. Relevant Therapeutic Options and New Potential Therapeutic Targets in Airway Remodeling

### 4.1. The Role of Standard Therapy in Airway Remodeling (LAMAs)

Prior randomized trials have shown that Inhaled Corticosteroids (ICS) can lead to a reduction in subepithelial fibrosis [155,156]. The use of Inhaled Corticosteroids/Long-Acting Beta2 Agonists (ICS/LABAs) is known to reduce airway inflammatory and remodeling pathways. For instance, in post-ICS-LABA treatment, a noted downregulation has been reported in the expression of various elements like nuclear receptor transcription coactivator, N-acetyltransferase, protein tyrosine kinase, nuclear receptor, and RNA polymerase II-activating transcription factor [157].

Muscarinic M1-3 receptors, present in the lungs, are crucial for the bronchodilatory effects of Long-Acting Muscarinic Antagonists (LAMAs), primarily induced through M3 inhibition. M3 receptors also influence mucus secretion, making LAMAs effective in reducing it. Muscarinic receptors are found in various lung cells, including epithelial cells, fibroblasts, smooth muscle cells, and inflammatory cells. This indicates that non-neuronal cells can also produce and release Acetylcholine (ACh), contributing to different biological responses in an autocrine or paracrine manner [158].

In animal and in vitro studies, LAMAs have shown significant anti-inflammatory and anti-proliferative effects. They are capable of inhibiting airway remodeling triggered by allergens [159].

ACh plays a role in airway inflammation and remodeling, also influencing the growth of ASM. Studies have shown the benefits of using muscarinic ACh Receptor (mAChR) antagonists, especially long-acting types [160] (LAMAs), to target these effects by blocking ACh’s activation of mAChRs.

In earlier stages of asthma, the challenge in prescribing LAMAs lies in the high variability of patient responses and the lack of detailed patient phenotyping. Enhancing the characterization of parasympathetic tone activity could lead to more effective LAMA prescriptions [161].

Adding LAMAs to ICS/LABA therapy enhances lung function, decreases exacerbation, and slightly improves asthma control in moderate to severe asthma patients not fully controlled by ICS/LABA alone. LAMAs are effective across various asthma phenotypes and endotypes. Three LAMA molecules—Tiotropium (TIO), Glycopyrronium (GLY), and umeclidinium—have been studied as add-ons, each with slightly different action onsets and half-lives. GLY, in particular, acts slightly faster than TIO, and umeclidinium may have similar properties [162].

The impact of anticholinergic drugs on airway remodeling remains unclear. Further research is needed to understand the anti-inflammatory effects of anti-muscarinic drugs on human airway inflammation and remodeling processes.

### 4.2. The Role of Biological Drugs in Airway Remodeling

The specific mechanisms of how environmental factors trigger the inflammatory responses leading to airway remodeling in asthma are not completely clear. Alarmins—cytokines from epithelial cells—start these immune processes, contributing to remodeling. Biological therapies can improve airflow by addressing inflammation and may reverse fixed remodeling caused by structural changes. Differentiating the immediate and long-term effects of biologics is vital for evaluating their impact on severe asthma’s airway remodeling [163].

Omalizumab, a humanized IgG1-κ monoclonal antibody, targets the Fc fragment of IgE [164]. It has been shown to reduce the thickness of the basement membrane and decrease fibronectin deposits in the airways of asthma patients [165].

Mepolizumab treatment in asthma patients has not only reduced the number of eosinophils in the bronchial passages, but has also decreased TGF-β1-positive eosinophils, the thickness and the tenascin immunoreactivity of the airways, and the levels of TGF-β1 in bronchoalveolar lavage fluid [166].

In biopsies from severe eosinophilic asthma patients, benralizumab significantly reduced eosinophils in the bronchial lamina propria and airway smooth muscle mass, without affecting myofibroblast numbers. This reduction was linked to the depletion of TGF-β1-positive eosinophils [167]. Additionally, a single dose of benralizumab notably improved ventilation in patients with uncontrolled asthma and significant mucus plugging [168].

In a mouse model of asthma, the use of dupilumab, which blocks both IL-4 and IL-13, was effective in preventing eosinophils from infiltrating lung tissue, though it did not impact the levels of circulating eosinophils [169]. In a different mouse model, blocking the IL-4Rα receptor improved lung function. This effect was achieved by influencing various factors involved in inflammation and the remodeling process in the lungs [170].

TSLP, which is overexpressed in asthmatic patients’ airway epithelia, activates lung fibroblasts, promoting airway remodeling [171]. Tezepelumab, a human IgG2-λ monoclonal antibody, targets TSLP. Studies show that TSLP’s role in fibrotic lung disease and its blockade reduce inflammation, TGF-β1 levels, and airway remodeling in animal models [172,173]. The CASCADE study revealed that Tezepelumab significantly reduces airway submucosal eosinophils in moderate-to-severe asthma patients compared to a placebo [174]. Lebrikizumab, a humanized monoclonal antibody, targets and inhibits soluble IL-13, blocking its downstream signaling. In exploratory analyses, treatment with lebrikizumab has been linked to a decrease in subepithelial fibrosis, a characteristic of airway remodeling [175].

The EMT in airway remodeling is influenced by the IL-33/CD146 axis. IL-33, derived from HDM extract-treated alveolar epithelial cells, stimulates CD146 expression. This process promotes EMT in the context of chronic allergic inflammation caused by HDM exposure. These findings highlight the potential of targeting the IL-33/CD146 pathway as a therapeutic approach towards airway remodeling [97].

### 4.3. Proteases as Potential Therapeutic Targets in Airway Remodeling

Regarding the targeting of allergen proteases as a potential therapeutic option, molecular allergology allows a precise diagnosis and optimal management of allergic asthma by employing allergen-specific immunotherapy as disease-modifying treatment.

To our knowledge, no therapeutic options specifically targeting allergen proteases are currently available. However, specific inhibitors of protease allergens have been considered as potential targets for therapeutic intervention in allergic diseases. Preclinical studies have shown that Der p 1-specific allergen delivery inhibitor compounds can prevent allergic sensitization and reduce inflammatory responses and clinical symptoms in asthma models [23,176]. Currently, no evidence is available in humans.

A phase II randomized controlled trial is evaluating the effects of a metalloprotease-12 inhibitor on allergen-induced airway responses, airway inflammation, and airway remodeling in subjects with mild eosinophilic HDM-allergic asthma (NCT03858686). Matrix metalloproteinases play a role in airway inflammation and remodeling; targeting endogenous and exogenous proteases could be a promising approach in the future.

## 5. Discussion

The loss of epithelial barrier function and airway remodeling are both features of allergic asthma.

The first one generally occurs in the early stages of the disease, commonly preceding the allergic sensitization. The second one has previously been considered to result from a long-lasting disease; however, evidence has shown that airway remodeling may occur in asthma patients even prior to diagnosis [93].

The barrier damage and the airway remodeling appear to be linked, defining a more complex clinical phenotype: epithelial permeability is higher in severe asthma compared with mild asthma [177], as well as in CRSwNP compared with CRS without nasal polyps [178].

Allergen proteases, such as Der p 1 and Asp f 13, have been shown to directly induce Airway Hyperresponsiveness (AHR) in animal models [179,180] and provoke morphological and molecular modifications in human ASM cells. These effects have been described not only in allergic patients, but also in subjects without prior allergic sensitization [87].

The complete inactivation of *Aspergillus* protease activity totally prevented T2 airway inflammation in a murine model of asthma [21]. Moreover, selective inhibition of Der p 1 not only reduced the levels of blood allergen-specific IgE, but also suppressed AHR in rats, avoiding chronic inflammation and the predisposition to airway remodeling [181].

The redox ambient in bronchial lumen regulates the response to allergen proteases. Der p 1 activity is enhanced by the bronchial epithelium-secreted glutathione-*S*-transferase-pi and by the presence of the antioxidant glutathione, both of which are highly present in human epithelial lung fluid [182]. In damaged epithelia of asthmatic patients, this effect is favored by anti-protease and mucociliary clearance impairment [183]. Moreover, in allergen protease-induced inflammation, the production of mitochondrial reactive oxygen species (ROS) is increased, feeding the inflammatory vicious circle. Downregulation of Indoleamine 2,3-dioxygenase (IDO) is observed in bronchial epithelial cells after exposure to *Aspergillus*, Der p 1, and HDM extracts [184,185].

Cystatin SN (CST1) inhibits cysteine protease activity, and its expression is enhanced in the epithelia of asthmatic patients. In a recent study, sputum and serum CST1 protein levels were negatively correlated with lung function in asthma patients; CST1 protein levels were significantly lower in the serum of HDM-specific IgE-positive asthmatic patients than in that of sIgE-negative asthmatic patients. Moreover, the HDM-induced epithelial barrier function disruption was suppressed by recombinant human CST1 protein in vitro and in vivo, reducing asthma symptoms. CST1 has been considered as a potential biomarker for monitoring asthma control [186].

The complex of different chemical and biological agents that humans are exposed to daily is known as the “exposome” [187] and includes microorganisms, pollution, hygiene-derived products, HDM, natural toxins, and food additives. The exposure to these factors alters cell function and favors the allergic response activation [188,189].

Proteases deriving from microorganisms, chemicals, environmental pollution, cigarette smoke, and other noxious agents may damage the epithelial barrier, similarly to allergen proteases, contributing to the inflammatory process, AHR, and airway remodeling [190].

Considering infective agents, the exposure to viruses during infancy and childhood predisposes humans to asthma development [191].

*Staphylococcus aureus* produces a wide range of proteins—including toxins, serine-protease-like proteins, and protein A—and its role in severe asthma and CRSwNP is well known. Staphylococcal enterotoxin B-IgE sensitization has been considered as a possible independent risk factor for asthma development and, in severe asthma patients, it has been associated with the presence of CRSwNP as a comorbidity [192]. Moreover, *Staphylococcus aureus*, regardless of enterotoxin production, may damage the airway epithelial cells, inducing the release of IL-25, IL-33, and TSLP, which can activate the ILC2 and the T2 response [193].

Recent evidence has demonstrated that, in the skin of a preclinical mouse model, eosinophil-recruiting chemokines (and eosinophil infiltration) are induced after *Staphylococcus aureus* epicutaneous exposure, and the IL-36α-IL-36R pathway is involved [194].

Exposure to tobacco smoke has been strongly associated with asthma prevalence in children [195] and exacerbates asthma and rhinitis symptoms in adults, decreasing muco-ciliary clearance [196]. Cigarette smoke can directly damage the TJ of the pulmonary epithelium [197], promoting the T2 response through epigenetic modifications such as decreasing the gene methylation of IL-4, IL-13 or increasing FOXP3 methylation after a HDM challenge [198,199].

Airborne microplastics inhalation caused pulmonary inflammatory cell infiltration and bronchoalveolar macrophage aggregation and also increased TNF-α levels in both healthy and asthmatic mice [200].

Air pollutants exacerbate the actions of aeroallergens, damaging the pollen cell wall and facilitating the release of allergenic proteins into the environment [201,202]. The direct allergenic protein–air pollutant contact promotes chemical protein modification before inhalation and deposition in the respiratory tract. In particular, high ozone (O_3_) and nitrogen dioxide (NO_2_) levels have been shown to efficiently nitrate and cross-link the proteins [203,204]. The concentrations of smog and industrial contamination-associated O_3_, NO_2_, and particles in suspension are geographically associated with higher rates of infant asthma [205]; also, maternal exposure to NO_2_ leads to enhanced sensitivity to allergens and increased AHR [206].

Increased T2 response and accumulation of ILC2 cells was observed in a diesel exhaust-enhanced allergic mice model [207]. O_3_ and NO_2_ promote the release of cytokines and chemokines, such as IL-33, IL-25, and TSLP, in both normal and asthmatic patients’ bronchial epithelial cells [208,209].

Recently, it has been observed that TLR4 is enhanced in Phl p 5, but not in Bet v 1 after ROS and nitrogen species exposure; subsequently, chemical modification and increased protein–receptor interactions occur. These events might contribute to the growing prevalence of respiratory allergies in industrialized countries [210].

Allergen proteases cause airway remodeling both directly and indirectly, through chronic inflammation-induced modifications.

HDM proteases directly induce the CX3CL1 chemokine, activating the T2 response [211] and the proliferation of ASMCs [212].

HDM proteases, signaling through the EGF receptor and TGF-β1, have been shown to promote epithelial-to-mesenchymal transition in human bronchial epithelium cells, contributing to airway remodeling in asthma [213].

Chronic inflammation induces a chronic repair reaction leading to the continuous release of growth factors; the uncontrolled proliferation of fibroblasts, ASMCs, and goblet cells; and the deposition of extracellular matrix molecules [214,215], which are all features of the airway remodeling process.

Due to the different emerging asthma phenotypes and the increasing number of factors included in the exposoma concept, which contributes to inflammatory damage and enhances asthma incidence and severity, the identification of clinical and radiological biomarkers in asthma is a concrete need. In a previous study, we explored the roles of the serum-free light chains -κ and -λ in asthma patients, showing their value as potential qualitative and quantitative (severity indicator) biomarkers, respectively [216].

Focusing on airway remodeling, the examination of the physiopathological and radiological features in allergic asthma should be considered for use in patient clusterization.

## 6. Conclusions and Future Directions

The role of allergen proteases in the pathogenesis of allergic asthma has been previously identified and is currently well known. The molecular mechanisms underlying allergen protease-induced damage, including epithelial barrier loss, have been better defined over the years and new allergen proteases have been identified.

In our review, the clinical features and the radiological patterns of airway remodeling have been explored, in order to emphasize the importance of biomarker identification in a disease with multiple endotypes and phenotypes.

The aim of this review was to draw a continuous thread between the molecular mechanisms of allergen protease exposure, epithelium damage, chronic inflammation, and airway remodeling in allergic asthma patients.

Although, as previously remarked, these “steps” are not necessarily subsequent, in many cases they may take part in an evolutive process. The lack of available biomarkers, in particular for monitoring airway inflammation and remodeling, does not allow the optimization of therapeutic management and follow-up for allergic asthma patients.

Considering T2 inflammation, beyond the blockage of IL-5, IL-4, IL-13, and IgE, the epithelial damage-derived cytokines are a newly available therapeutic target.

In the future, better endophenotyping of asthmatic patients will ensure the selection of the appropriate therapeutic option, utilizing the increasing number of available drugs. Currently, different available molecules have shown positive effects on airway remodeling. However, the identification of new potential therapeutic targets in the molecular pathways involved in the airway remodeling process should be achieved, considering the increase of allergic, environmental, and chemical stimuli in the industrialized exposome.

## Figures and Tables

**Figure 1 ijms-25-05747-f001:**
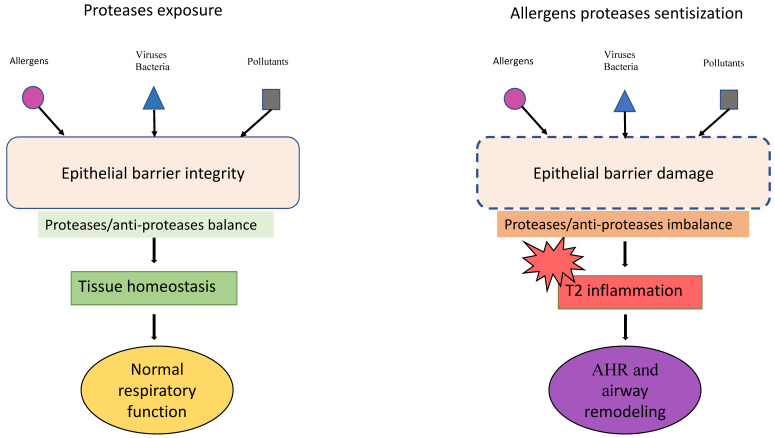
The role of allergen proteases in epithelial barrier damage, chronic inflammation, and airway remodeling. AHR: Airway Hyperresponsiveness.

**Figure 2 ijms-25-05747-f002:**
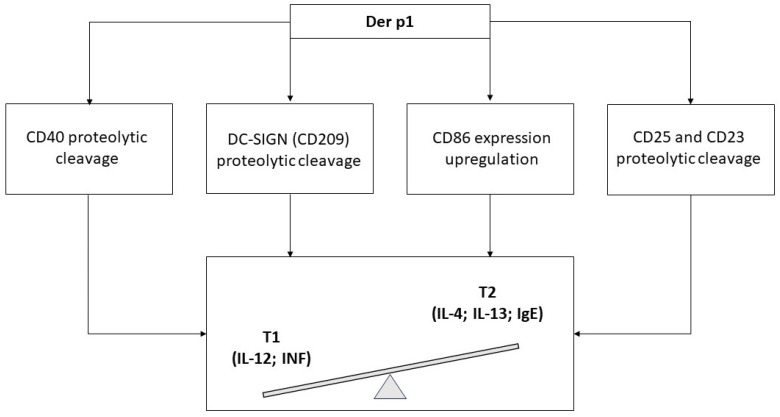
Der p 1 effects on CD system in promoting T2 immune response.

**Table 1 ijms-25-05747-t001:** Radiological markers of airway remodeling.

Marker	Description	References
High-Resolution CT (HRCT)	HRCT is crucial for identifying static and dynamic airway changes in asthma, revealing details as small as 1 mm in diameter.	[122,123]
Bronchial Wall Thickness (% WT)	% WT, the bronchial-to-arterial diameter ratio (BA ratio), and the level of airway collapsibility (AC) are acknowledged as efficient measurements for assessing airway remodeling in CT scans.	[124]
Wall Area Percentage (WA%)	WA% is a crucial marker for assessing airway remodeling in severe asthma, with a negative correlation between WA% and FEV1 observed, indicating the relationship between airway wall thickness and lung function impairment.	[126]
Quantitative CT (qCT) Scans	QCT scans serve as effective markers for airway remodeling, enhancing the precise analysis of severe asthma. Biomarkers such as wall thickness percentage (WT%), wall area percentage (WA%), and air trapping are higher in asthma patients and are particularly elevated in severe cases.	[92,127,128,129]
Bronchial Wall Thickness (BWT) and Emphysema	BWT and emphysema are more prevalent in patients with severe asthma, indicating their roles as radiological markers for lung function changes caused by asthma.	[125,133,134,135]

**Table 2 ijms-25-05747-t002:** Biomarkers of airway remodeling.

Biomarker	Description	References
Sub-Epithelial Fibrosis	Characterized by thicker airway smooth muscle, mucous gland hyperplasia, angiogenesis, and damaged epithelial layers, contributing to stiffer airway walls.	[91,92]
Epithelial Remodeling	Involves deterioration of epithelial cells, loss of ciliated cells, and an increase in goblet cells. The epithelial–mesenchymal transition (EMT) driven by TGF-β is a key process, with markers like reduced E-cadherin and increased N-cadherin.	[94,95,96]
Reticular Basement Membrane (RBM) Thickening	Linked to gene expressions affecting airway growth and fibrosis. The identification of specific fibrocytes in BALF as markers suggests a role in airway remodeling.	[104,105]
Subepithelial Fibrosis	TGFβ‘s role in transforming airway fibroblasts into myofibroblasts leads to subepithelial fibrosis. The severity of fibrosis correlates with TGFB1 mRNA levels, and periostin’s association with IL-4 and IL-13 impacts fibrosis and inflammation.	[107,108,110]
Airway Smooth Muscle (ASM)	ASM cell mitogens, such as PDGF, TGFβ, and EGF, are involved in asthma. Histology assessed through endobronchial biopsies serves as a valuable biomarker.	[112,113]
Mucus	Hypersecretion of mucins MUC5AC and MUC5B by goblet cells contributes to airway remodeling; targeting MUC5AC secretion could be a potential therapeutic strategy.	[119,120]

## Data Availability

The data used in the study are available upon reasonable request to the corresponding author.

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
