# Peer review of "The New Paradigm: The Role of Proteins and Triggers in the Evolution of Allergic Asthma"

_ijms, 2024, doi:10.3390/ijms25115747_

Round 1

Reviewer 1 Report

Comments and Suggestions for Authors

The proposed objective of this review is to give an overview on the role of proteases in allergy mechanisms, as stated also in the Conclusions: "The aim of this review was to draw a continuous thread between molecular mechanisms of allergen protease exposure, epithelium damage, chronic inflammation and airway remodeling in allergic asthma patients." Nevertheless, the focus is sometimes difficult to follow along the text because the subject is very wide if treated from the molecular level to the medical-therapeutic-social level. More specifically:

- what is this "new paradigma" mentioned in the title? Tracking proteases to establish the context of a certain allergic case?

 - Fig. 1 is not fully clear: what is the meaning of the "Allergens proteases action" arrow between the healthy tissue and the damaged tissue? These should be the two distinct situations that may occur in different patients. Moreover, if the substrates of allergenic proteases are the membrane proteins on cell surface and at the junctions, do proteases act from the extracellular portion of the membrane proteins? Where would the protease inhibitors exert their protective action? These aspects are not shown in the Figure, not described in the text.

- page 5: a scheme showing the mechanisms involving proteases and CD systems should be added.

- what is the reason to keep Table 1 and Table 2 separated? For example, QCT is reported among markers but it is defined as a biomarker. What do authors exactly mean with markers and biomarkers?

- pages 9-10 report medical aspects that look a bit out-of-focus with respect to the topic of the review.

- in general, the paragraphs are too fragmented in many sentences separated by full-stop marks, which look disconnected.

Reviewer 2 Report

Comments and Suggestions for Authors

Baglivo et al provide a review which start with the discussion of the role of allergens proteases, followed by a airway remodeling topic, and finishing with biomarkers and relevant therapeutic options. As I was reading the review, I felt confused with the flow of the topics. The authors spent a large part of the review discussing the mechanisms by which proteases can contribute to asthma. However, the topic is forgotten when the authors discuss the biomarkers and therapeutic options. Can proteases be used as biomarkers for the diagnosis of asthma? Are there clinical trials that target proteases as a treatment for asthma? 

Furthermore, the authors provide only one figure. Perhaps making others which demonstrate the role of proteases in the asthma microenvironment or how targeting them could benefit the patients would contribute for a better understanding of the authors perspective.

Comments on the Quality of English Language

The english language needs to be improved to allow a better flow of the reading. The paragraphs organization is confusing.

Round 2

Reviewer 2 Report

Comments and Suggestions for Authors

Dear authors,

After the revisions, I believe the manuscript has improved. However, I still think that a quick organization with the paragraphs would contribute for a better reading of the review. Nonetheless, it looks good for publication in my opinion.

Comments on the Quality of English Language

N/A

Author Response

Dear authors,

After the revisions, I believe the manuscript has improved. However, I still think that a quick organization with the paragraphs would contribute for a better reading of the review. Nonetheless, it looks good for publication in my opinion.

Thank you for your insightful comments and suggestions regarding our manuscript.

Our rationale is that the broader medical aspects included are crucial for providing a comprehensive understanding of the topic. We believe that these sections contribute significantly to the context and depth of the review, linking molecular mechanisms to broader diagnostic and therapeutic strategies which are essential for a holistic understanding of allergic asthma.

We feel that the inclusion of these elements will benefit readers by presenting a fuller picture of the complexities involved in the field; however, we accepted your suggestion and the titles of some paragraphs (tracked in the manuscript) have been modified, and the tables have been moved, in order to give more continuity in reading of our review.  

Thank you once again for your careful reading and valuable critiques. We are grateful for the opportunity to improve our manuscript and look forward to your decision.